# Comparative Study of Green and Traditional Routes for Cellulose Extraction from a Sugarcane By-Product

**DOI:** 10.3390/polym15051251

**Published:** 2023-03-01

**Authors:** Francisca Casanova, Ricardo Freixo, Carla F. Pereira, Alessandra B. Ribeiro, Eduardo M. Costa, Manuela E. Pintado, Óscar L. Ramos

**Affiliations:** CBQF—Centro de Biotecnologia e Química Fina—Laboratório Associado, Escola Superior de Biotecnologia, Universidade Católica Portuguesa, Rua Diogo Botelho 1327, 4169-005 Porto, Portugal

**Keywords:** green chemistry, auto-hydrolysis, deep eutectic solvents, green metrics, cellulose, sugarcane bagasse

## Abstract

Sugarcane bagasse (SCB) is the main residue of the sugarcane industry and a promising renewable and sustainable lignocellulosic material. The cellulose component of SCB, present at 40–50%, can be used to produce value-added products for various applications. Herein, we present a comprehensive and comparative study of green and traditional approaches for cellulose extraction from the by-product SCB. Green methods of extraction (deep eutectic solvents, organosolv, and hydrothermal processing) were compared to traditional methods (acid and alkaline hydrolyses). The impact of the treatments was evaluated by considering the extract yield, chemical profile, and structural properties. In addition, an evaluation of the sustainability aspects of the most promising cellulose extraction methods was performed. Among the proposed methods, autohydrolysis was the most promising approach in cellulose extraction, yielding 63.5% of a solid fraction with ca. 70% cellulose. The solid fraction showed a crystallinity index of 60.4% and typical cellulose functional groups. This approach was demonstrated to be environmentally friendly, as indicated by the green metrics assessed (E(nvironmental)-factor = 0.30 and Process Mass Intensity (PMI) = 20.5). Autohydrolysis was shown to be the most cost-effective and sustainable approach for the extraction of a cellulose-rich extract from SCB, which is extremely relevant for aiming the valorization of the most abundant by-product of the sugarcane industry.

## 1. Introduction

Cellulose, the world’s most abundant natural polymer and a promising renewable and biodegradable material, can be used to produce valuable cellulose-based products [1,2,3]. The most exploited natural source of cellulose is wood [4,5,6], but non-wood plant fibers [7,8,9] represent cellulose sources with great potential and increased demand. In the context of sustainable development, lignocellulosic biomass from industrial and agricultural wastes has attracted much attention as cellulose sources. These bio-residues are highly available, have low costs, contribute to solving disposal problems for industries, and their use permits the increase in the value of underutilized renewable materials [8,10]. Among agricultural crops, sugarcane plantations for the sugar and alcohol industries are famous for their volumes and large amounts of residues. Sugarcane bagasse (SCB) is the main residue of the sugarcane industry, representing almost 30% (by weight) of the sugarcane agricultural product, reaching a production of 300 million tons per year [11,12]. Currently, it is burned for energy production and/or used for low-value applications (such as animal feed). Thus the development of new cellulose-based added-value products is an excellent route to promote its valorization, being of extreme importance for both economic and environmental reasons [13,14].

In SCB, the cellulose fibrils, the main polymer found in this residue (40–50%), are embedded in a matrix mainly composed of hemicellulose (25–35%) and lignin (18–24%) interacting with other molecules such as polyphenols. The cellulose extraction from lignocellulosic biomass has been the object of intensive research to meet the industrial goals of accomplishing the sustainable development goals. In the literature, it is possible to find cellulose extraction methods from different plant materials besides SCB, namely corn stover, soybean, banana leaves, rice husk, cocoa pod husk, and acai berry, among others [15,16,17,18], with the ultimate goal of the obtention of industrially relevant products in a wide range of applications such as the textile, pharmaceutical, and food industries, besides its biomedical applications and the development of cellulose-based biopolymers [16,17,19]. A wide range of physical (e.g., ball-mill process, microwave, extrusion, and ultrasonication); chemical (e.g., acid or alkaline hydrolyses, organosolv process, and deep eutectic solvents); physicochemical (e.g., steam explosion), and biological methods (whole-cell treatment or enzymatic pretreatment) have been proposed for cellulose extraction and purification [19,20,21,22,23]. Treatments consisting of acid [24,25] and alkali [26,27] hydrolyses have been the most widely employed for cellulose extraction from SCB. However, they pose problems regarding environmental pollution and safety (harsh toxic chemicals are used, which generate large amounts of residues). Moreover, they require expensive resistant equipment and considerable waste treatment [21,28]. Physical methods, such as microwave [29,30], ultrasonic [31], and physicochemical methods as steam explosion [32,33], can also be utilized, but they represent a significative equipment acquisition investment, have a high-energy consumption (economic and environmental impact), and have scalability challenges. Furthermore, biological treatments such as enzymatic [34] are high-cost, time-consuming, and only moderately efficient [28]. With sustainable development becoming the top of the agenda, researchers are continuously looking for greener, safer, and efficient methods for cellulose extraction. Methods that have recently attracted considerable attention include extraction with deep eutectic solvents and through hydrothermal processes.

Deep eutectic solvents (DESs) are a novel class of sustainable solvents that are currently subject to extensive research [35]. They consist of an hydrogen bond acceptor—HBA (e.g., quaternary ammonium salts) and hydrogen bond donor(s)—HBD (e.g., amines, amides, carboxylic acids), which result in a depletion of melting temperature [36,37]. DESs exhibit the advantages of ionic liquids: chemical and thermal stability, non-flammability, high dissolution ability, and good ionic conductivity. However, they have a lower cost and are simple to prepare, non-toxic, and often biodegradable, making them very attractive from a green chemistry perspective [38]. Moreover, these solvents exhibit tunable properties depending on their composition, thanks to the uncountable possibilities of combinations of HBA and HBD. Even so, these solvents present some disadvantages, such as difficult separation from the reaction products and complex design of the solvent to be used (HBA and HDB selection, as well as their ratio) [39,40]. DESs have been used for the treatment of several lignocellulosic materials [41,42,43], including SCB [44,45,46], where the most commonly used compounds have been choline chloride (an available, cheap, non-toxic, and biodegradable compound), urea, and lactic acid.

Hydrothermal processing is a promising clean technology to fraction lignocellulosic materials into added-value compounds [47]. In this process, liquid water at high temperatures (150–230 °C) and pressures (P > Psat) fraction materials through hydronium-catalyzed reactions by the hydronium ions generated from water ionization—autohydrolysis [48,49]. Autohydrolysis is considered an environmentally friendly process since it does not require the addition and recovery of chemical reagents, has a low by-product generation, has limited equipment corrosion problems, and is simple to operate, making it one of the main choices for industrial applications [47,50,51]. Furthermore, this process enables a high recovery of cellulose through hemicellulose depolymerization and lignin degradation, consequently increasing the cellulose availability [47,48]. This process has been applied to various lignocellulosic materials [52,53], including SCB [49,54]. Organosolvent-assisted hydrothermal processing involves the use of aqueous solutions of organic solvents (e.g., methanol, ethanol, acetone), which partially hydrolyze lignin and lignin–carbohydrate bonds, resulting in a solid residue composed mainly of cellulose [55]. Some of its advantages are sulfur-free emissions and a simple recovery process, as the organic solvents can be easily recycled by distillation [56,57]. This process has been applied to wood [58], non-wood fibers [59], and agricultural wastes [60,61], including SCB [62]. Recent articles have demonstrated the potential of using sequential treatments to remove hemicellulose and lignin. Details on the combination of auto-hydrolysis followed by organosolv delignification processes of wood [63,64] and agro-industry wastes [60,65,66], namely SCB [56,67], have been published.

Even though in the literature there are a few isolated articles exploring green processes for cellulose extraction, there is no available information showing a comparative study of the various methods, both green and traditional, with the same feedstock material. Articles exploring the use of DESs [44], hydrothermal treatment [68,69], and organosolv [70] for the extraction of cellulose from SCB exist, but a detailed comparative study, including an integrated analysis of extract yields, chemical composition, and structural characterization through different and complementary techniques is lacking.

The use of an efficient, low cost, and eco-friendly extraction process to obtain cellulose from a complex matrix, such as sugarcane bagasse, constitutes one of the key technical challenges found by biorefineries in the previous years [71]. Herein, we present a comprehensive study of green and traditional routes for cellulose extraction from the most abundant by-product of the sugarcane industry, aiming to provide valuable knowledge and information to permit supported decisions on SCB valorization. Promising green methods, i.e., DESs and organosolv and hydrothermal processing, were tested and compared to traditional acid and alkaline hydrolyses. The impact of the different treatments was evaluated regarding product yield, cellulose content, and lignin and hemicellulose removal. A detailed solid-state structural characterization by a variety of techniques (Fourier Transform Infrared Spectroscopy (FT-IR), Powder X-ray Diffraction (PXRD), and Scanning Electron Microscopy (SEM)) was performed for all approaches studied. An applicability analysis based on sustainability (E(nvironmental)-factor and Process Mass Intensity (PMI)) and energy consumptions was also performed for the most promising processes, taking this article even further than others of its kind. As per our knowledge, this is the first report where greener approaches are compared with traditional cellulose extraction methods for the same SCB feedstock material, and where an integrated analysis of results is performed focused on the product yields, structural characterizations, and sustainability evaluations, which is very useful for planning biorefineries based on this waste.

## 2. Materials and Methods

### 2.1. Reagents

The reagents used during the hydrolysis process were of an analytical grade or higher. Sulfuric acid (95.0%) and ethanol (99.8%) were purchased from Honeywell Fluka; sodium hydroxide (97.0%), choline chloride (98.0%), and cellulose > 95.0% were purchased from Sigma-Aldrich; and urea (99.5%) and lactic acid (90.0%) were purchased from Merck.

### 2.2. Raw Materials

Sugarcane bagasse (SCB) harvested in June 2019 was supplied by Raízen (Araçatuba, Brazil). SCB exhibiting 36–40% moisture content was dried at 40 °C until reaching a moisture content of ca. 5–10%. Then, the SCB was ground in a blade mill (Retsch Cutting Mill SM 100) using a 4 mm sieve. The 4 mm milled bagasse was used in the comparative screening of traditional and green methods for cellulose extraction. A part of the milled bagasse was sieved in a vibratory sieve shaker (Retsch AS 200 basic) and the fraction with a particle size > 315 µm was used in the auto-hydrolysis optimization. A schematic representation of the methodology used for SCB pretreatment and cellulose extraction is presented in Figure 1.

### 2.3. Acid and Alkaline Hydrolyses

Acid hydrolysis (AcH) was conducted according to Guerra-Rodríguez [72], with some modifications. Briefly, a sulfuric acid solution prepared at 2% (*w*/*w*) was added to the SCB at a 12:1 ratio (*w*/*w*). Then, the solution was heated in a water bath (Julabo SW 22) at 100 °C for 30 min. The solid fraction obtained after filtration was washed with ca. 3 L water for each 10 g of biomass until neutrality (i.e., pH 6–7), and then dried at 55 °C overnight (ca. 10–12 h) in a thermostatic incubator (VWR INCU-Line 150R).

The alkaline hydrolysis (AlkH) was conducted according to Wen Wang [73] with some modifications. Briefly, a sodium hydroxide solution prepared at 2% (*w*/*v*) was preheated to 85 °C and added to SCB at a ratio of 20:1 (*v*/*w*). Then, the solution was heated in a water bath at 85 °C for 60 min. The solid fraction obtained after filtration was washed with ca. 3 L water for each 10 g of biomass until neutrality (i.e., pH 6–7), and then dried at 55 °C overnight (ca. 10–12 h) in a thermostatic incubator (VWR INCU-Line 150R).

### 2.4. Deep Eutectic Solvents (DESs)

In this study, deep eutectic solvents (DESs) prepared from choline chloride as the hydrogen bond acceptor (HBA) and urea or lactic acid as the hydrogen bond donor (HBD) were evaluated for the obtention of cellulose-rich materials from SCB. Each solvent was prepared by mixing choline chloride and its corresponding hydrogen bond donor in the specified ratio. The mix was carried out at 80 °C for 1–2 h until a clear liquid was formed. The different DESs used and the HBA to HBD ratios are listed in Table 1.

After the DES preparation, 500 mg of SCB was added to 20 g of each DES formulation, and the dissolution of biomass was carried out at 80 °C in a water bath with agitation 130 rpm for 24 h. The reaction mixtures were then cooled down to room temperature (ca. 20 °C) and 50 mL of distilled water was added. The undissolved biomass was separated by centrifugation, washed with deionized water, and then dried at 55 °C in an oven for 24 h.

### 2.5. Hydrothermal Treatments

The hydrothermal treatments were carried out in a 3.75 L reactor (Parr Instruments Company, Moline, IL, USA). For each batch, 100 g of SCB was placed in the reactor and supplemented with the proper amount of tap water (sample AuH) or ethanol at 50% (*v*/*v*) (ethanol-assisted hydrothermal treatment—sample EtOH), in order to set a liquid to solid ratio of 15:1, with a temperature of 170, 180 or 190 °C for water and 170 °C for ethanol for 1 h. After the hydrolysis stage was completed, the reactor was cooled to room temperature and the treated samples were filtered through a cotton strainer in a beaker. For sample AuH_EtOH, a second successive treatment was tested after the auto-hydrolysis process, with a solution of ethanol 50% (*v*/*v*) in a liquid to solid ratio of 15:1 and with a temperature of 190 °C for 2.5 h.

### 2.6. Chemical Composition

The laboratory analytical procedure (LAP) for standard biomass analysis of the National Renewable Energy Laboratory (NREL) was used for the determination of structural carbohydrates, lignin, and ash in biomass [74,75]. The total carbohydrate content in the filtrate solution was quantified by a high-performance liquid chromatograph (HPLC) equipped with a refractive index detector and an Aminex HPX 87H column 300 × 7.8 mm (Bio-rad laboratories, Hercules, CA, USA). The chromatograms were run in isocratic mode at 0.6 mL/min and 50 °C. The mobile phase employed was 0.027% (*v*/*v*) sulfuric acid solution and the injection volume was 10 µL. All samples were analyzed at least in duplicate.

### 2.7. Fourier Transform Infrared Spectroscopy (FT-IR) 

The FT-IR spectra were recorded using the Frontier™ MIR/FIR spectrometer from PerkinElmer in a scanning range of 550–4000 cm^−1^ for 32 scans at a spectral resolution of 4 cm^−1^. All analyses were conducted at least in duplicate.

### 2.8. Powder X-ray Diffraction (PXRD)

Powder X-ray Diffraction (PXRD) analyses were performed on a Rigaku MiniFlex 600 diffractometer with Cu kα radiation, with a voltage of 40 kV, and a current of 15 mA (3° ≤ 2θ ≥ 60°; step of 0.01 and speed rate of 3.0°/min). The Segal crystallinity index (CI, %) was calculated using the following equation:(1)$$\mathrm{CI}\left(\%\right)=\frac{I_{t}-I_{a}}{I_{a}}\times 100$$
where *I_t_* is the total intensity of the (200) peak for cellulose *I* at 22.5° 2θ, and *I_a_* is the amorphous intensity at 18° 2θ for cellulose *I* [76,77]. 

### 2.9. Scanning Electron Microscopy (SEM)

The morphology of the samples was evaluated by scanning electron microscopy (SEM) on JSM-5600 LV Scanning Electron Microscope from JEOL, Japan. Prior to analysis, the samples were placed in observation stubs covered with double-sided adhesive carbon tape (NEM tape, Nisshin, Japan) and coated with Au/Pd using a Sputter Coater (Polaron, Bad Schwalbach, Germany). All observations were performed in a high-vacuum with an acceleration voltage of 20 kV, at a working distance of 9 or 10 mm and a spot size of 4. The images presented here are representative images of the morphology of each sample.

### 2.10. Sustainability Evaluation

In order to measure the environmental impact of the most promising cellulose extraction processes, the E(nvironmental)-factor (E(factor)), the Process Mass Intensity (PMI), and the energy consumption [78,79,80] have been evaluated according to the Equations (2)–(4), respectively:(2)$$E\left(factor\right)=\frac{Dry\;mass\;of\;waste\;\left(\mathrm{kg}\right)}{Mass\;of\;final\;product\;\left(\mathrm{kg}\right)}$$
(3)$$\mathrm{PMI}=\frac{Mass\;of\;consumables\;\left(\mathrm{kg}\right)}{Mass\;of\;final\;product\;\left(\mathrm{kg}\right)}$$
(4)$$\mathrm{Energy}\;\mathrm{consumption}\;\left(\mathrm{kWh}/\mathrm{kg}\right)=\frac{Electric\;energy\;consumption\;of\;the\;equipment\;\left(\mathrm{kWh}\right)}{Mass\;of\;final\;product\;\left(\mathrm{kg}\right)}$$

In the PMI metric, the water is considered as a consumable, and in the energy consumption (kWh/kg), the electric energy consumption was estimated using a Perel power meter. 

### 2.11. Statistical Analysis

All analyses were performed at least in duplicate, and the results are reported as the mean values and standard deviations. One-way analysis of variance (ANOVA) followed by post-hoc Tukey’s test (*p* < 0.05) were conducted to determine the significant differences between the mean values using SPSS 28.0 Statistical Software Program (SPSS Inc., Chicago, IL, USA).

## 3. Results and Discussion

### 3.1. Comparative Screening of Traditional and Green Methods for Cellulose Extraction

This section presents an integrative discussion of the traditional (acid and alkaline hydrolyses) and greener extraction methods (using deep eutectic solvents, hydrothermal processing, and organosolv processing) for cellulose from SCB, encompassing the process yield, chemical composition, and structural characterization of the resultant extracts.

#### 3.1.1. Chemical Composition

The SCB presents cellulose as the main component (i.e., 36.2%), followed by lignin—acid soluble and Klason lignin (acid insoluble lignin)—and hemicellulose (i.e., 24.1 and 22.6%, respectively) (Table 2). This result agrees with previously reported data for this kind of biomass, which showed a composition of 39.9% cellulose, 22.7% hemicellulose, and 24.2% lignin [81]. The difference found in the percentage of cellulose may be due to different factors, such as the type of soil or climate conditions (e.g., temperature, relative humidity, irrigation), where sugarcane was planted [82].

An acid hydrolysis applied to SCB allowed the purification of cellulose, which resulted in a statistically significant increase (*p* < 0.05) in the cellulose percentage (i.e., 42.5%), in relation to the content of cellulose in SCB (i.e., 36.2%), with the remaining components (i.e., hemicellulose and lignin) not being statistically significantly affected (*p* > 0.05) in relation to the chemical composition (see Table 2). This result showed that the acid hydrolysis was not successful in removing hemicellulose and lignin. The same was already described for wheat straw, where lignin was not effectively removed [69]. Other studies conducted with the same concentration of sulfuric acid and time of reaction, but with a higher temperature, 150 °C instead of 100 °C applied here, achieved 56.5% of cellulose in the extracted biomass. This difference may be attributed to the higher temperature applied, since it is well known that the temperature and time of reaction have significant impacts on the extraction of cellulose [25]. The utilization of higher temperatures in the reaction also implies an increase in energy input, which can lead to the degradation of pentose sugars to furfural and other undesirable compounds. Moreover, the use of acids presents a risk for corrosion of the equipment and leads to the generation of large amounts or residues that requires treatment [25]. The alkaline hydrolysis was shown to be the most statistically effective approach (*p* < 0.05) in cellulose extraction in comparison to acid hydrolysis, once it allowed a significant increase in cellulose percentage (*p* < 0.05) (i.e., 57.1%) and the removal of both soluble and Klason lignin contents, with this removal being statistically significant (*p* < 0.05) for Klason lignin (i.e., 11.4% and 12.8% in relation to the SCB and AcH samples, respectively). This approach appeared to be the best in the delignification of SCB (i.e., lignin removal of 51.9 %). This result is in agreement with the findings of Maryana and co-workers [26], which showed a lignin removal of ca. 59%, but applying double the amount of sodium hydroxide, i.e., 4% (*w*/*v*), to SCB. It is well established that the alkaline hydrolysis with sodium hydroxide allows for the breakdown of the alpha-aryl ester bonds from the polyphenolic monomers, partially decomposing lignin while solubilizing hemicellulose and lignin through hydrogen bond weakening, increasing the pulp purity in cellulose [83].

From DESs applied to cellulose extraction, the combination of choline chloride and lactic acid as HBA and HBD solvents, respectively, were shown to be the most promising approach to obtain a solid fraction (i.e., yield of 90.7%) with the lowest content of hemicellulose (17.7%) and soluble lignin (5.9%). In a recent study conducted by Satlewal et al. [84], using the same combination of HBA and HBD, the authors showed a higher removal of lignin, achieving a delignification of 50%. Although the DES process has been reported as a cost-effective approach to fractionate lignocellulosic material, it had the lowest efficiency in obtaining a solid fraction rich in cellulose from SCB in comparison with acid (AcH sample) or the alkaline (AakH sample) hydrolysis, since it produces the poorest fraction in cellulose (i.e., 34.7%)—see Table 2. Among these methods, the DES was the most efficient in the removal of hemicellulose (i.e., a statistically significant reduction of 4.9% in relation to SCB), while the alkaline hydrolysis was the most efficient in the removal of lignin (i.e., statistically significant reduction of 1.1% in soluble lignin and 11.4% in Klason lignin in relation to SCB), leading to the richest fraction in cellulose, i.e., 57.1% (Table 2). In relation to all processes tested, autohydrolysis was the best, achieving the highest percentage of cellulose, with a statistically significant increase (*p* < 0.05) of 22.9% from bagasse and the highest reduction of hemicellulose with a statistically significant decrease (*p* < 0.05) of 11.2% from bagasse. In agreement with our results, Santucci et al.’s [85] work also showed a high percentage of hemicellulose (10%) removal with auto-hydrolysis treatment in sugarcane bagasse at 170 °C for 1 h. This behavior is likely due to the high temperatures applied during the heating phase in a reactor that leads to the cleavage of the acetyl linkages by water and acetate or other acids, such as acetic and uronic acids, from hemicelluloses when they are released. This release catalyzes the hydrolysis and promotes the removal of oligosaccharides and sugar hemicelluloses [47].

The biorefinery concept is one of the most important current scientific challenges, through biotechnological, chemical, and thermochemical processes, for obtaining more than one value-added product [13]. This concept could be applied to lignocellulosic material through the auto-hydrolysis process, since it enables the separation and recovery of hemicellulose in the liquid fraction, when high temperatures are applied, from the solid fraction composed of cellulose and lignin. Then, through the application of a second treatment, such as alkaline hydrolysis, which is efficient in lignin removal, it would be possible to obtain a liquid fraction rich in lignin and a solid fraction rich in cellulose. This pathway can be applied to agri-food wastes and by-products as sugarcane bagasse to prompt the main components of the lignocellulose biomass, drastically reducing the waste generated and the negative impact to the environment, while promoting its valorization in high-quality fractions.

#### 3.1.2. Structural Characterization

All samples were fully characterized in terms of functional groups and crystallinity using FT-IR and PXRD, respectively (Figure 2).

The FT-IR transmittance spectra of SCB, cellulose fractions extracted through different approaches, and commercial cellulose >95% (CC) are shown in Figure 2a. The vibration bands corresponding to the typical functional groups for cellulose, lignin, and hemicellulose, as well as the samples where these peaks were detected, are listed in Appendix A. The characteristic cellulose vibrations, which are clear in the CC spectrum (Figure 2a), have also been found in all samples, as shown in Figure 2a, namely at: 3200–3400 cm^−1^, 2900 cm^−1^, 1600–1640 cm^−1^, 1430 cm^−1^, 1110 cm^−1^, and 900 cm^−1^, indicating that cellulose was preserved after all extraction processes. Similar vibrations in the wavelength positions observed here have been reported in the literature for cellulose fractions extracted from SCB by both traditional (alkaline hydrolysis) [7,26] and green (DES) methods [45,46]. The presence of a vibration band at 1500 cm^−1^ (Figure 2a) visible in all samples, except for CC, indicates the presence of lignin, which is in agreement with the results of the chemical composition presented in Table 2. The same has been observed in the literature for cellulose extracts obtained from SCB through green methods, such as organosolv [8], steam explosion [69], and DES [46], where lignin absorption bands are still present. A complete absence of these bands, as observed on the CC sample, was already achieved for alkaline hydrolysis [86] when more severe conditions were applied (17.5% (*w*/*v*) sodium hydroxide for 3 h).

In the present work, differences in the FT-IR transmittance spectra can be observed for the different extraction methods. The absence of the vibration bands at 1730 cm^−1^ and 1250 cm^−1^ in the alkaline and auto-hydrolysis samples (Figure 2a) indicates the partial removal of hemicellulose and lignin, while for the remaining samples, the removal did not appear to be as effective (these vibrations are still prominent). In fact, the alkaline treated sample was the only one, where the peak at 830 cm^−1^ (associated with lignin) was absent, while the auto-hydrolysis sample was the only treatment where the peak at 2850 cm^−1^ (associated with hemicellulose) was absent (Figure 2a). These results are in agreement with the chemical composition discussed before (Table 2), as the alkaline hydrolysis sample showed the least amount of lignin, while the auto-hydrolysis sample presented the least amount of hemicellulose. Furthermore, the presence of a vibration band at 1060 cm^−1^ for the auto-hydrolysis sample allied to the increase in the intensity of the band at 1110 cm^−1^ (attributed to C-O-C ring stretching in cellulose), as well as the increase in the band intensity at 1600 cm^−1^ for the alkali-treated sample, indicate that these treatments were more efficient in producing a fraction purer in cellulose, as these bands have been reported as characteristic for cellulose functional groups [7,87], which are clearly present in the CC sample (electronic Appendix A). These results corroborate the chemical profiles presented earlier (Table 2), as alkaline and auto-hydrolysis fractions showed the highest cellulose content.

The crystallinity of SCB, cellulose fractions and commercial cellulose, were analyzed by powder X-ray diffraction. The PXRD patterns and the crystallinity indices are shown in Figure 2b. All samples exhibited the characteristic cellulose reflections at 2θ = 15.0°/16.5°, 22.5°, and 34.5°, corresponding to the (1–10)/(110), (200), and (004) planes of cellulose I, respectively [7,88]. The crystallinity index (CI) of SCB was 38.2 ± 0.4% and increased significantly (*p* < 0.05), i.e., CI ranging from 56.1 to 61.0%, depending on the extraction process applied, due to the removal of lignin and hemicellulose species as amorphous parts of the extracts. This increase has also been reported in the literature, namely for hydrothermal [68] and DES [46] treatments of sugarcane bagasse. However, for the hydrothermal processing, the reported increase was of only ca. 5%, compared to 17.9% observed here; and for alkaline hydrolysis, the CI has even been reported to decrease, compared to the increase of 21.8% achieved here, demonstrating an added-value to the processes herein developed [26]. An increase in cellulose content after extraction was corroborated by the chemical profiles of the extracted fractions (Table 2) and the presence of cellulose functional groups (Figure 2a). Nonetheless, the fact that these samples presented statistically significantly lower (*p* < 0.05) crystallinity index values than the commercial cellulose (i.e., CI of 79.7%, Figure 2b) suggests the presence of residual amorphous parts, such as lignin and hemicellulose, results that are in agreement with chemical composition analysis and the FT-IR spectra (Table 2 and Figure 2a, respectively). The additional reflections on the alkaline-hydrolyzed sample and in the sample treated with DES suggest the presence of residual impurities such as salts deriving from the extraction processes.

### 3.2. Optimization of the Hydrothermal Treatment

Taking into consideration that the hydrothermal process in the Parr reactor was the most promising method for obtaining a solid fraction richer in cellulose based on the green approaches tested, it was decided to evaluate the impact of different temperatures (i.e., 170, 180, and 190 °C) to obtain a purer fraction of sugarcane bagasse. So, the fraction with size > 315 µm, representing ca. 70% of the total bagasse, was selected after the milling process because it is purer in cellulose, presenting a higher cellulose content (44.8% compared to 36.2% in the total fractions of bagasse, Table 3 and Table 2 respectively) and a significantly lower percentage of ashes (1.6% compared to 4.4% in the total fractions of bagasse—information determined but not shown here). Higher temperatures (180 and 190 °C) than the initial tested (i.e., 170 °C) for auto-hydrolysis, as well as ethanol-assisted methods, were tested in an attempt to increase the extraction of cellulose, since higher temperatures or the use of ethanol helps to hydrolyze lignin bonds and lignin–carbohydrate bonds, promoting a higher removal of these components, which leads to a fraction purer in cellulose [55,85,89].

#### 3.2.1. Chemical Composition

Using SCB fraction with particle size > 315 µm, the auto-hydrolysis process at 170 °C achieved a higher purification of cellulose fraction (i.e., 69.4% of cellulose content and a hemicellulose removal of 26.1%—Table 3), when compared with the cellulose fraction obtained from all size fractions of bagasse (i.e., 59.1% of cellulose content and a hemicellulose removal of 10.9%—Table 2), likely due to the higher level of impurities as hemicellulose, lignin, and ashes. Since auto-hydrolysis led to better results in terms of the cellulose fraction with a higher purity, higher temperatures of hydrolysis were studied, i.e., 180 and 190 °C, in order to evaluate if its increase potentiates the removal of contaminants (i.e., lignin and hemicellulose) in accordance with what has been published in the literature [56]. From the results, it was possible to observe that the increase in heating temperature from 170 to 180 and 190 °C did not significantly impact (*p* > 0.05) the cellulose purity, as well as the hemicellulose removal. In another study, it was possible to find that only temperatures above 160 °C potentiates the removal of hemicellulose, and when the difference between the temperatures applied are low, no impact is observed in the hemicellulose removal [85]. The increase from 170 °C to 190 °C showed only an increase of 9% in hemicellulose removal with no statistical significance (*p* > 0.05). Another approach that has been widely reported in the literature as a promising green process to achieve fractions with a higher purity of cellulose from lignocellulosic biomass comprises the use of organosolv [89]. In this way, two processes encompassing an organosolv treatment and an association of two treatments, i.e., auto-hydrolysis followed by an organosolv, were studied in a Parr reactor. In the first process, a solution of ethanol at 50% (*v*/*v*) tested at 170 °C for 1 h was compared in terms of its performance to the second treatment, comprising the association of both processes, firstly an auto-hydrolysis at 170 °C for 1 h followed by an organosolv with ethanol at 50% (*v*/*v*), at 190 °C for 2.5 h. The results showed that the second process comprising the association of both treatments led to the production of a faction with a statistically significantly higher (*p* < 0.05) content in cellulose (i.e., 80.3%) in relation to the first process comprising an organosolv treatment, but statistically significant differences were not observed (*p* > 0.05) in relation to auto-hydrolysis treatment applied at 170 °C (AuH_170 sample)—Table 3. In fact, statistically significant differences were also not obtained (*p* > 0.05) for the removal of hemicellulose and Klason lignin in relation to auto-hydrolysis at 170 °C (AuH_170 sample)—Table 3. In terms of an energetic point-of-view, considering the solvents required, the residues generated, and the complexity, the association of the two processes represents a more complex, expensive, and time-consuming extraction process, since this reaction required more than double the processing time and the use of ethanol, while the auto-hydrolysis at 170 °C was less costly and complex. In addition to this, the yield of process in terms of solid fraction production for the association of treatments was much lower compared to all others, with the auto-hydrolysis at 170 °C and organosolv being the processes that showed the highest yield (i.e., 63.5 and 66.3%, respectively). From all conditions tested, the auto-hydrolysis process at 170 °C, using only water and a lower heating temperature in the Parr reactor, was the most promising approach in terms of process yield, cellulose level, and percentage of contaminants (i.e., lignin and hemicellulose)—see Table 3.

#### 3.2.2. Structural Characterization

Cellulose fractions obtained by hydrothermal processing of the SCB fraction with particle size superior to 315 µm were characterized in terms of functional groups and crystallinity using FT-IR and PXRD, respectively (Figure 3).

The FT-IR spectra of the SCB fraction superior to 315 µm, cellulose fractions extracted through different approaches, and commercial cellulose >95% are shown in Figure 3a. The characteristic cellulose absorption bands have been found in all samples, as shown in Figure 3a (Appendix A), namely at: 3200–3400 cm^−1^, 2900 cm^−1^, 1600–1640 cm^−1^, 1430 cm^−1^, 1110 cm^−1^, and 900 cm^−1^, indicating that cellulose was preserved after all treatments. Almost all extracted fractions (with the exception of the fraction treated with ethanol at 50% (*v*/*v*) at 170 °C for 1 h–i.e., AuH_EtOH sample) presented defined cellulose characteristic vibrations at 1060 cm^−1^ and 1110 cm^−1^, as opposed to SCB (Figure 3a) and the samples discussed in Section 3.1.2, indicating an improvement in cellulose isolation. Furthermore, the absence of the vibration bands at 2850 cm^−1^ and 1730 cm^−1^ (except for the AuH_EtOH sample) and the decrease in the intensity of vibration at 1250 cm^−1^ in all samples, when compared to SCB (Figure 3a), indicate an efficient removal of hemicellulose and lignin species beyond the removal observed when using SCB of all sizes. Untreated SCB and the ethanol treated fraction (AuH_EtOH sample) did not show prominent characteristic cellulose bands at 1060 cm^−1^ and 1110 cm^−1^, and in contrast, showed vibration bands at 1730 cm^−1^ and 1250 cm^−1^ (associated with lignin and hemicellulose) (Figure 3a), which is in agreement with the chemical analysis results, as these showed a lower cellulose content and considerable lignin presence (Table 3). The use of a fraction of SCB superior to 315 µm as the starting material allowed us to obtain fractions richer in cellulose when using auto-hydrolysis (170–190 °C) and auto-hydrolysis followed by organosolv with ethanol, as seen by the FT-IR spectra (Figure 3a) and chemical analysis (Table 3), when compared with the results of SCB of all sizes (Figure 2a and Table 2). This primarily has to do with the highest cellulose content of the starting material (due to the lower presence of small impurities such as ash), since the increase in the percentage of cellulose after processing was revealed to be similar using the two starting materials (Table 3).

The CI of the SCB fraction superior to 315 µm, cellulose fractions extracted through different approaches, and commercial cellulose > 95% (CC) were analyzed by PXRD. The PXRD patterns and the CI are shown in Figure 3b. All samples exhibited the characteristic cellulose reflections as previously described (i.e., reflections at 2θ = 15.0°/16.5°, 22.5°, and 34.5°, corresponding to the (1–10)/(110), (200), and (004) planes of cellulose I, respectively) [7,88]. The CI of the SCB fraction superior to 315 µm (i.e., 46.2%, Figure 3b) was significantly higher (*p* < 0.05) than the one calculated for the total SCB (i.e., 38.2%, Figure 2b), which is in agreement with the higher cellulose content determined by chemical analysis. This CI was significantly increased (*p* < 0.05) after all extraction processes (CI ranging from 57.3 to 70.1%, depending on the extraction process applied), where the association of auto-hydrolysis with organosolv processes (AuH_EtOH sample) exhibited a statistically higher (*p* < 0.05) crystallinity (i.e., CI of 70.1%), followed by the auto-hydrolysis processes (CI ranged from 60.4 to 60.6%, depending on the heating temperature applied)—Figure 3b. These results were corroborated by chemical composition analysis, which showed that the combination of auto-hydrolysis with organosolv presented the highest (*p* < 0.05) cellulose content, followed by the auto-hydrolysis process at 170 °C (Table 3). The CI herein calculated is higher than the ones reported in the literature for fractions extracted from sugarcane bagasse through hydrothermal processes, i.e., about 40% [68]. This indicates that the processes explored here were able to produce a solid fraction richer in cellulose, exhibiting a higher crystallinity. This is likely due to the fact that the processes were more efficient in the removal of contaminant residues as hemicellulose and lignin.

SCB and the extracted fibers through the combination of auto-hydrolysis with organosolv and the auto-hydrolysis process at 170 °C, which were the most promising processes in obtaining a solid fraction purer in cellulose, were evaluated in terms of their morphology by SEM analysis (Figure 4), in order to determine the differences in fiber surface and structure after the extraction through these processes.

Figure 4a shows the morphology of untreated SCB fibers for a fraction superior to 315 µm, which showed a compacted and agglomerated structure. The surface of these fibers presented a smooth surface, which may be due to the presence of lipids and impurities (e.g., waxes, pectin, oils), as described by Saelee et al. and Kumar et al. [7,69]. During the hydrolysis processes, the temperature and pressure seem to have led to the defibrillation and depolymerization of SCB, resulting in separated and delaminated fibers (Figure 4b,c). Figure 4b,c also shows a rougher fiber surface, probably as a result of the removal of surface impurities, such as lipids, hemicelluloses, and lignin. These results show a noticeable consistency with the compositional analysis and FT-IR and PXRD data. Similar changes in morphology have been reported in the literature for SCB treated by hydrothermal processing [69], organosolv [8], and DES [46].

### 3.3. Sustainability and Cost Considerations

The increased interest in green and sustainable development is driving a transition from a traditional and linear flow of materials to the integration of green chemistry and circular economy principles into the development, manufacturing, and commercialization of materials and products [78]. The processing of the sugarcane feedstock and the by-products within the sugarcane industry is of utmost importance for obtaining more than one added-value product from vegetal biomass (e.g., bioenergy and bio-based products) in an integrated valorization approach. This falls under the biorefinery concept, which is a key enabling strategy for a circular economy and one of the main current scientific challenges [13,14]. The use of a SCB feedstock that is not only a renewable biomass, but also involves the valorization of unavoidable waste biomass, makes these processes green and sustainable from a feedstock perspective. Even so, the use of green and sustainable processes, involving waste minimization and avoiding the use of toxic and/or hazardous reagents and solvents, is also highly desirable.

To properly measure the environmental impact of chemical processes, dedicated assessment tools have been described, allowing for the comparison of the greenness of different extraction methods. Different factors characterized by a different level of complexity are currently used to assess the environmental impact of chemical processes [79]. The E(nvironmental)-factor and the Process Mass Intensity (PMI) are the preferred metrics for evaluating the environmental impact of a chemical process, as they are fairly easy to generate and communicate [90]. A higher E-factor implies a higher amount of waste generated and, consequently, a greater negative environmental impact. The closer to zero the value of E-factor is, the less waste generated and more sustainable and greener the process will be [91]. A lower PMI value (closer to 1) means fewer resources are needed to make 1 kg of product, and, hence, the more efficient and less impactful the process is [90]. Taking into consideration the results shown before, the 2 most promising processes in extracting a solid fraction rich in cellulose from SCB (i.e., auto-hydrolysis treatment applied at 170 °C for 1 h—AuH_170, and the association of auto-hydrolysis at 170 °C for 1 h with organosolv with ethanol at 50% (*v*/*v*) at 190 °C for 2.5 h—AuH_EtOH) were evaluated and compared in terms of their greenness: E-factor, PMI, and energy consumption (Table 4).

From the results above, we can conclude that, from an E-factor perspective, both processes developed are considerably sustainable, as E-factors can be between 1 and 5 for bulk chemicals, between 5 and 50 for fine chemicals, and between 25 and 100 for pharmaceuticals [78,92,93,94]. The auto-hydrolysis process is more sustainable than the ethanol-assisted process, as would be expected by the additional use of an organic solvent. The PMI value is still lower than traditional fine chemical processes (40–100) [92] for both processes; however, the values were not as close to 1 as desirable, showing that the process could be screened for further improvements. Even so, auto-hydrolysis proved to be more sustainable than the ethanol-assisted process, and it should be noted that, even though the consumables for this process represent a considerable mass per kilogram of product, these consumables are a by-product of the sugar industry and a renewable agricultural waste (i.e., SCB) and water, which is one of the most sustainable solvents and one that can be potentially recovered and recycled from wastewater [95]. Regarding energy consumption, the auto-hydrolysis process requires less energy than the ethanol-assisted one, as would be expected considering the latter is a 2-step process with a longer processing time (i.e., 2 h of reaction). References in the literature regarding energy consumption in cellulose extraction are rare or often very vague, but values of 30 kwh/kg have been reported for mechanical processes [96]. It should be noted that these values refer to industrial scale processes, and that the approaches here developed were at the bench scale. If a similar scale is considered, probably significantly lower values of energy consumption could be obtained by the processes herein described.

Regarding costs, lower E-factors have been shown to correlate well with reduced manufacturing costs, a reflection of lower process materials inputs and outputs, reduced costs of hazardous and toxic waste disposal, improved capacity utilization, and reduced energy demand [78]. As the E-factor is higher for the ethanol-assisted process when compared to auto-hydrolysis, this process is expected to have a higher cost impact. This can be explained by the fact that it is a two-step process with a longer process time, meaning higher energy consumption and time costs. In addition, it uses an organic solvent, resulting in higher resources and additional waste management costs. Furthermore, this process has a lower product yield, which is not beneficial in terms of industrial and cost considerations.

## 4. Conclusions

The sugarcane bagasse, an underutilized agriculture residue, can be used as feedstock to obtain a solid fraction rich in cellulose through a sustainable and eco-friendly method. In this work, SCB fraction superior to 315 µm was revealed to be the most suitable fraction for cellulose extraction since it presents a higher content in cellulose and a lower content in small impurities. Taking into consideration all the results obtained through the different and complementary techniques employed (i.e., chemical analysis, FT-IR, PXRD, and SEM characterization, but also product yield and sustainability considerations/energy consumption), the autohydrolysis process at 170 °C for 1 h was revealed to be the most promising cellulose extraction process, while also being the most attractive approach under an economic point-of-view. This process proved to be sustainable (E-factor of 0.30 and PMI of 20.5) and effective at the cellulose extraction and hemicellulose removal, producing a cellulose-rich fraction from SCB (69.4 ± 0.4) with a considerable yield (63.5%), exhibiting the typical cellulose functional groups and a crystallinity index of 60.4 ± 0.9%. For a greater lignin removal, additional treatments are demanded, such as bleaching treatments, in order to achieve a purer and whiter cellulose fraction. This may allow for its use in more refined applications, such as cosmetics or pharmaceuticals, where the cellulose content and purity are essential.

## Figures and Tables

**Figure 1 polymers-15-01251-f001:**
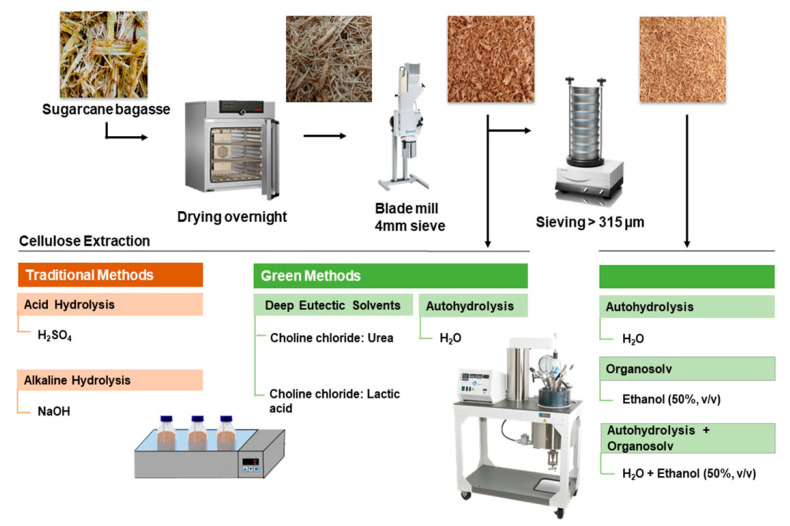
Schematic representation of sugarcane bagasse pretreatment and cellulose extraction methods tested.

**Figure 2 polymers-15-01251-f002:**
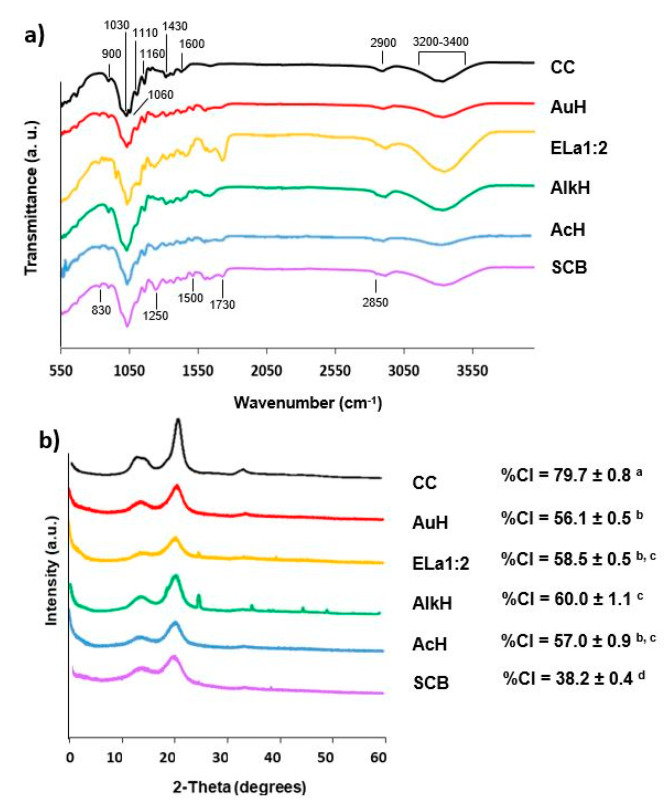
Structural characterization by (**a**) Fourier transform infrared spectrometry and (**b**) powder X-ray diffraction analysis of untreated sugarcane bagasse (SCB) and fractions extracted by: auto-hydrolysis (AuH), DES with choline chloride and lactic acid (ratio 1:2) (Ela1:2); alkaline hydrolysis (AlkH); and acid hydrolysis (AcH), in comparison with commercial cellulose (CC) used as control. CI averages followed by the same letters do not differ statistically by Tukey’s test (*p* < 0.05).

**Figure 3 polymers-15-01251-f003:**
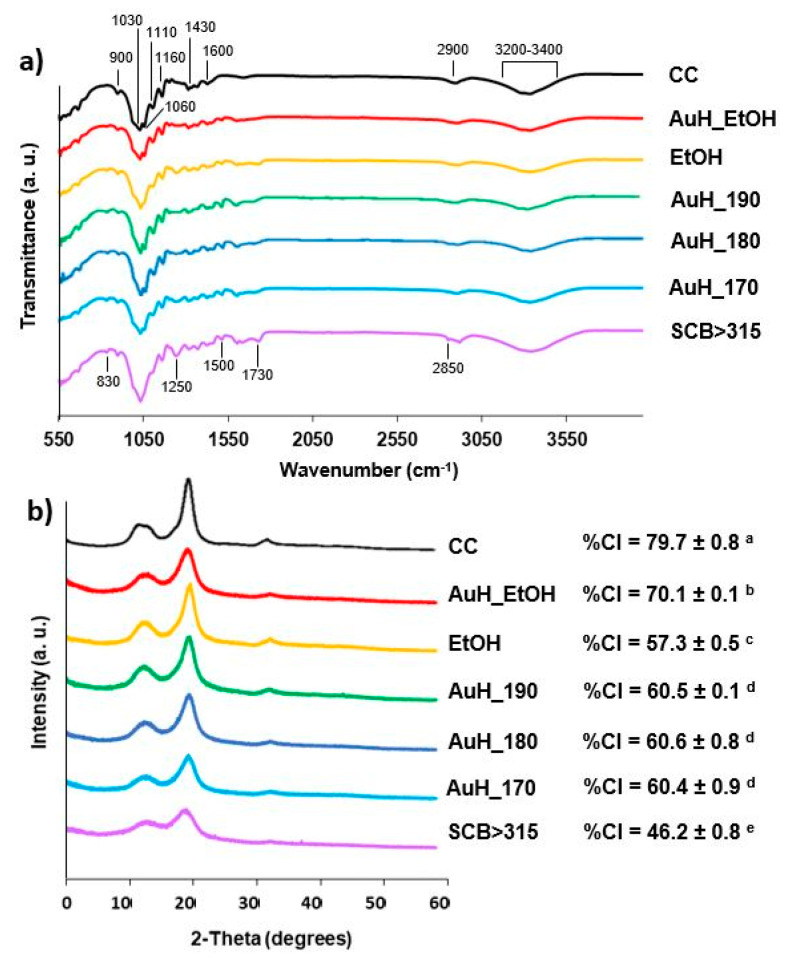
Structural characterization by (**a**) Fourier transform infrared spectrometry and (**b**) powder X-ray diffraction analysis of untreated sugarcane bagasse fraction superior to 315 µm (SCB > 315), and fractions extracted by: auto-hydrolysis followed by organosolv with ethanol (AuH_EtOH); organosolv with ethanol (EtOH); auto-hydrolysis at 190 °C (AuH_190); auto-hydrolysis at 180 °C (AuH_180); and auto-hydrolysis at 170 °C (AuH_170), in comparison with commercial cellulose (CC) used as control. CI averages followed by the same letters do not differ statistically by Tukey’s test (*p*  <  0.05).

**Figure 4 polymers-15-01251-f004:**
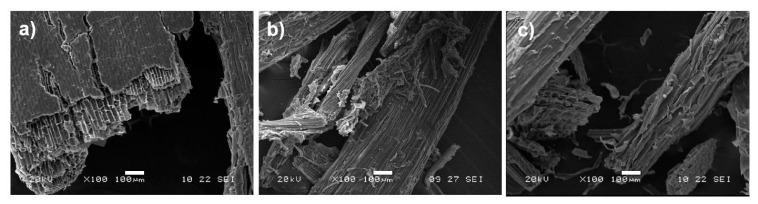
SEM images of the (**a**) untreated SCB fibers for a fraction superior to 315 µm; (**b**) auto-hydrolysis followed by organosolv with ethanol (AuH_EtOH) and (**c**) auto-hydrolysis (AuH_170) treated SCB fibers.

**Table 1 polymers-15-01251-t001:** Conditions of the cellulose extraction from sugarcane bagasse using deep eutectic solvents.

Sample	HBA	HBD	Ratio HBA:HBD
EU1:1	Choline chloride	Urea	1:1
EU1:2			1:2
ELa1:2		Lactic acid	1:2

Legend: EU1:1: DES ratio choline chloride:urea 1:1; EU1:2 DES ratio choline chloride:urea 1:2; Ela1:2: DES ratio choline chloride:latic acid:1:2.

**Table 2 polymers-15-01251-t002:** Yield and chemical composition (in terms of percentage of cellulose, hemicellulose, and soluble and Klason lignin) of the total fraction of SCB and solid fractions rich in cellulose obtained from different methods.

	Yield (%)	Chemical Composition (%)			
		Cellulose	Hemicellulose	Soluble Lignin	Klason Lignin
SCB—all fractions *	-	36.2 ± 1.1 ^a^	22.6 ± 0.6 ^a,b^	4.1 ± 0.2 ^a^	20.0 ± 0.3 ^a^
AcH	84.5	42.5 ± 0.4 ^b^	23.9 ± 0.3 ^a,b^	3.8 ± 0.0 ^a,e^	21.4 ± 0.3 ^a^
AlkH	63.9	57.1 ± 0.5 ^c^	19.9 ± 0.1 ^a,c^	3.0 ± 0.0 ^a,e^	8.6 ± 0.6 ^b^
EU1:1	90.5	27.8 ± 0.3 ^d^	20.6 ± 0.4 ^a,c^	9.2 ± 0.2 ^b^	10.9 ± 0.4 ^b^
EU1:2	92.7	34.3 ± 1.0 ^a^	25.9 ± 1.8 ^b^	11.4 ± 0.3 ^c^	15.5 ± 1.0 ^c^
ELa1:2	90.7	34.7 ± 0.9 ^a^	17.7 ± 0.3 ^c^	5.9 ± 0.2 ^d^	15.8 ± 0.1 ^c^
AuH	56.2	59.1 ± 0.9 ^c^	11.4 ± 0.2 ^d^	2.3 ± 0.6 ^e^	26.0 ± 0.0 ^d^

Legend: * Sugarcane bagasse all size fractions; AcH: Acid hydrolysis; AlkH: Alkaline hydrolysis; EU1:1: DES ratio choline chloride:urea 1:1; EU1:2 DES ratio choline chloride:urea 1:2; Ela1:2: DES ratio choline chloride:latic acid 1:2; AuH: Auto-hydrolysis. Averages followed by the same letters in the same column do not differ statistically by Tukey’s test (*p* < 0.05).

**Table 3 polymers-15-01251-t003:** Yield and chemical composition (in terms of percentage of cellulose, hemicellulose, and soluble and Klason lignin) of the cellulose-rich fractions obtained from the SCB fraction superior to 315 µm.

	Yield (%)	Chemical Composition (%)			
		Cellulose	Hemicellulose	Soluble Lignin	Klason Lignin
SCB > 315 µm *	-	44.8 ± 2.1 ^a^	34.9 ± 7.8 ^a^	5.3 ± 0.1 ^a^	23.5 ± 0.0 ^a^
AuH_170	63.5	69.4 ± 0.4 ^b,c^	8.8 ± 1.6 ^b^	2.4 ± 0.3 ^b^	24.9 ± 2.3 ^a,b,c^
AuH_180	53.2	63.0 ± 1.6 ^b^	11.3 ± 1.4 ^b^	2.0 ± 0.0 ^c^	30.8 ± 0.0 ^a,b^
AuH_190	49.4	64.4 ± 2.1 ^b^	7.9 ± 1.5 ^b^	2.3 ± 0.0 ^b, c^	32.0 ± 0.0 ^b^
EtOH	66.3	61.1 ± 3.1 ^b^	13.2 ± 2.3 ^b^	4.8 ± 0.0 ^a^	21.3 ± 2.6 ^a,c^
AuH_EtOH	42.4	80.3 ± 4.2 ^c^	0.00 ± 0.0 ^b^	1.9 ± 0.1 ^c^	17.0 ± 1.2 ^c^

Legend: * Fraction of Sugarcane bagasse superior to 315 µm; AuH_170: auto-hydrolysis at 170 °C for 1 h; AuH_180: auto-hydrolysis at 180 °C for 1 h; AuH_190: auto-hydrolysis at 190 °C for 1 h; EtOH: organosolv with ethanol at 50% (*v*/*v*) at 170 °C for 1 h; AuH_EtOH: auto-hydrolysis at 170 °C for 1 h followed by organosolv with ethanol at 50% (*v*/*v*) at 170 °C for 1 h. Averages followed by the same letters do not differ statistically by Tukey’s test (*p* < 0.05).

**Table 4 polymers-15-01251-t004:** E-factor, PMI, and energy consumption values for the auto-hydrolysis cellulose extraction process (170 °C, 1 h) and auto-hydrolysis (170 °C, 1 h) associated with organosolv with ethanol 50% (*v*/*v*) (190 °C, 2.5 h) from SCB (fraction > 315 µm).

Process	E-factor	PMI	Energy Consumption (kWh/kg)
AuH_170	0.30	20.5	15.2
AuH_EtOH	0.56	49.33	23.2

## Data Availability

The datasets used and/or analyzed during the current study are available from the corresponding author on reasonable request.

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
