# Peer review of "Comparative Study of Green and Traditional Routes for Cellulose Extraction from a Sugarcane By-Product"

_polymers, 2023, doi:10.3390/polym15051251_

Round 1
Reviewer 1 Report
Some of my comment of the work under review are given below.
1. How the author define green extraction. elaborate more please.
2. Pleas more precisely the advantage of the green extraction. How they compared the extracted yield.
3. The SEM mrciograph are not conclusive, please add more conclusive images.
4. The author should add 13CMR spectra of the final purified extract of cellulose.
5. please avoid discussion literature in the conclusion. summarize your important results in it.
6. How the sustainability and cost of the method was ascertain. please add the formula for further reading.
Author Response
"Dear Reviewer, please see the attachment".

Reviewer 2 Report
The experimental article "Comparative Study of Green and Traditional Routes for Cellulose Extraction from a Sugarcane by-product" is fully consistent with the theme of the publication "Polymers". The work is characterized by high relevance in view of the study of various methods of pre-treatment on the by-product of sugar cane. The title favorably emphasizes the content of the article and will attract a wide range of readers.
The strengths of the work are the choice of the actual object of study - sugarcane bagasse and the comparison of various types of pre-treatment, followed by a detailed assessment of the final product.
The weak side of the work is reflected below in the comments, recommendations and questions of the reviewer:
1. Delete text on lines 229-231.
2. Clause 3.1. duplicates the meaning of the introduction. It needs to be rephrased and returned to the introduction.
3. On lines 253-258 it is written that acid hydrolysis led to an increase in the mass fraction of cellulose from 36.2% to 42.5%, but the mass fractions of lignin and gemmicellulose did not change. Explain due to which components the mass fraction of cellulose increased?
4 The authors write "soluble and Klason lignin" throughout the text, it would be more successful to speak "acid-soluble and acid-insoluble lignin".
5. Why were short treatments (1 h) with highly dilute solutions (2%) of acid (H2SO4) and (NaOH) chosen as traditional methods?
Reviewer 3 Report
In this manuscript, aiming the utilization of sugarcane bagasse, authors extracted cellulose by green and traditional methods from the residue of the sugarcane industry. The effect of different parameters on the extract efficiency have been investigated. In general, the manuscript is well organized. However, there are still some issues to be addressed. A minor revision is required before its acceptance.
1. The contents from the template should be deleted, for example, the contents between section 3 and section 3.1.
2. The tables should be expressed in three-line table.
3. Some of the figures should be modified with better readability, especially the texts.
4. More background on the cellulose extraction should be further clarified with supporting articles: Utilization of Acai Berry Residual Biomass for Extraction of Lignocellulosic Byproducts; A mixed acid methodology to produce thermally stable cellulose nanocrystal at high yield using phosphoric acid; Characterization of natural fiber from manau rattan (Calamus manan) as a potential reinforcement for polymer-based composites; etc.
5. The table A1 is suggested to be shifted into the supporting information.
6. There are still some typos and grammar issues to be corrected.
Round 2
Reviewer 1 Report
Accept in the present form